# Development of Low-Grade Serous Ovarian Carcinoma from Benign Ovarian Serous Cystadenoma Cells

**DOI:** 10.3390/cancers14061506

**Published:** 2022-03-15

**Authors:** Puja Dey, Kentaro Nakayama, Sultana Razia, Masako Ishikawa, Tomoka Ishibashi, Hitomi Yamashita, Kosuke Kanno, Seiya Sato, Tohru Kiyono, Satoru Kyo

**Affiliations:** 1Department of Obstetrics and Gynecology, Shimane University Faculty of Medicine, Izumo 693-8501, Japan; puja1001066@gmail.com (P.D.); raeedahmed@yahoo.com (S.R.); m-ishi@med.shimane-u.ac.jp (M.I.); tomoka@med.shimane-u.ac.jp (T.I.); meme1103@med.shimane-u.ac.jp (H.Y.); kanno39@med.shimane-u.ac.jp (K.K.); sseiya@med.shimane-u.ac.jp (S.S.); satoruky@med.shimane-u.ac.jp (S.K.); 2Project for Prevention of HPV-Related Cancer, National Cancer Center, Exploratory Oncology Research and Clinical Trial Center (EPOC), Kashiwa 277-8577, Japan

**Keywords:** carcinogenesis, low-grade serous ovarian carcinoma, ovarian serous cystadenoma, ovarian cancer, *KRAS*, *PIK3CA*

## Abstract

**Simple Summary:**

Low-grade serous ovarian carcinoma (LGSOC) is thought to progress from benign cystadenoma in a stepwise fashion via serous borderline tumors (SBTs). This hypothesis is based on pathological and molecular evidence obtained following the genetic analysis of clinical samples from LGSOCs, SBTs, and cystadenomas. However, there have been no reports on the occurrence of LGSOCs following the introduction of oncogenes into benign serous cystadenoma cells. This study successfully developed an in vitro carcinogenic model of LGSOCs by introducing oncogenic *KRAS* and *PIK3CA* gene mutations in immortalized HOVs-cyst-1 cells from serous cystadenomas. The established mouse xenograft tumors resulting from the inoculation of HOVs-cyst-1 cells with *KRAS* and *PIK3CA* mutations exhibited the micropapillary invasive pattern of LGSOCs with low nuclear atypia without alveoli.

**Abstract:**

Despite the knowledge about numerous genetic mutations essential for the progression of low-grade serous ovarian carcinoma (LGSOC), the specific combination of mutations required remains unclear. Here, we aimed to recognize the oncogenic mutations responsible for the stepwise development of LGSOC using immortalized HOVs-cyst-1 cells, developed from ovarian serous cystadenoma cells, and immortalized via *cyclin D1*, *CDK4^R24C^*, and *hTERT* gene transfection. Furthermore, oncogenic mutations, *KRAS* and *PIK3CA*, were individually and simultaneously introduced in immortalized HOV-cyst-1 cells. Cell functions were subsequently analyzed via in vitro assays. *KRAS* or *PIK3CA* double mutant HOV-cyst-1 cells exhibited higher cell proliferation and migration capacity than the wild-type cells, or those with either a *KRAS* or a *PIK3CA* mutation, indicating that these mutations play a causative role in LGSOC tumorigenesis. Moreover, *KRAS* and *PIK3CA* double mutants gained tumorigenic potential in nude mice, whereas the cells with a single mutant exhibited no signs of tumorigenicity. Furthermore, the transformation of HOV-cyst-1 cells with *KRAS* and *PIK3CA* mutants resulted in the development of tumors that were grossly and histologically similar to human LGSOCs. These findings suggest that simultaneous activation of the KRAS/ERK and PIK3CA/AKT signaling pathways is essential for LGSOC development.

## 1. Introduction

Ovarian cancer, of which the serous type is the most frequently encountered, is reportedly the most lethal gynecologic malignancy [1,2]. Additionally, ovarian neoplasms are heterogeneous and can be classified as type I or type II neoplasms, which are characterized by distinct molecular, histopathological, and clinical features [3,4]. Type II ovarian serous neoplasms, which predominantly include high-grade serous carcinomas (HGSCs) with *TP53* mutations in approximately 96% of cases, show an aggressive clinical course [5], whereas type I tumors include low-grade serous ovarian carcinomas (LGSOCs), mucinous carcinomas, clear cell carcinomas, and endometroid ovarian carcinomas. Specifically, LGSOCs constitute a relatively unusual yet distinctive type of tumor that tends to occur in younger patients, have indolent progression and long-term survival, and exhibit a stronger association with chemoresistance than HGSCs [6]. Several studies have demonstrated that in Western countries, the development of LGSOC results from either *KRAS* (16–54%) or *BRAF* (2–33%) mutations [7,8,9], whereas the frequencies of *PIK3CA* (4.8%) and *ERBB2* (9.5%) mutations, in this context, are very low [8,10]. Thus, in Western countries, the KRAS/BRAF/ERK signaling pathway is thought to play an essential role in developing LGSOCs. However, the molecular profile of LGSOCs in Japanese patients has not yet been sufficiently investigated. In a previous study, we reported a Japanese case of LGSOC with synchronous serous adenofibroma, atypical proliferative serous tumor (APST), and noninvasive micropapillary serous borderline tumor (MPSC) [11]. The coexistence of serous adenofibroma, APST, MPSC, and LGSOC, in this case, suggests that LGSOCs develop from serous adenofibroma in a stepwise fashion. However, sequence analysis of *KRAS* or *BRAF* genes in different pathological regions revealed no oncogenic mutations, suggesting that these mutations may not contribute to tumor progression in Japanese cases of LGSOCs [11]. In a subsequent study, we observed a high frequency of *PIK3CA* mutations (60%) despite the absence of *KRAS* mutation in Japanese cases of LGSOCs [12]. This implies that the primary carcinogenic signaling pathways underlying LGSOCs in Japanese and Western populations may differ. Further, activation of the PIK3CA/AKT signaling pathway may play a significant role in the carcinogenesis of Japanese LGSOCs. Concurrent *KRAS/BRAF* and *PIK3CA* mutations have also been found in SBTs and LGSOCs [10,12].

Regardless of the evidence that has been accumulating on the prevalence of these molecular alterations, the specific combination of genetic mutations required for the progression of LGSOC has not yet been examined sufficiently. Therefore, a suitable research model to identify the molecular mechanism underlying the development of LGSOC is urgently needed. Previous observations, including ours, prompted us to investigate the minimal genetic requirements for LGSOC development [7,8,9,10,11,12]. Thus, we generated immortalized epithelial cells from ovarian cystadenoma cells and introduced oncogenic *KRAS, PIK3CA,* and *KRAS + PIK3CA* mutations, which appeared to be the most essential for developing LGSOCs. We conducted experiments involving a xenograft mouse model to evaluate the carcinogenic potential of these gene mutations. After that, we analyzed the proliferative and metastatic behavior of cells carrying these mutations via in vitro assays. The in vitro carcinogenic model established in this study can enhance the current understanding regarding the molecular pathogenesis of LGSOC while also contributing to the development of novel therapeutic agents for the management of LGSOCs.

## 2. Materials and Methods

### 2.1. Purification and Isolation of Ovarian Serous Cystadenoma Epithelial Cells

A human ovarian serous cystadenoma tissue sample was obtained via laparoscopic bilateral salpingo-oophorectomy from a 53-year-old postmenopausal woman in June 2016. The resected specimen was reviewed by a gynecological pathologist (N.I.) and confirmed as ovarian serous cystadenoma. All research protocols were approved by the Institutional Research Ethics Committee of Shimane University (IRB No. 20070305-2, version 10; last update, 8 December 2019). Additionally, informed consent was obtained from the patient. To confirm that the malignant lesions were not mixed with the collected samples, we examined the tissues for pathological outcome and ensured that there was no malignant tissue. A 10 cm tissue culture dish was used to collect serous cystadenoma tissue samples (5 cm); the sample was cleaned with antiseptic phosphate-buffered saline (PBS, Gibco, Carlsbad, CA, USA) solution. To clearly visualize epithelial cells (Figure 1a), the tissue was cut lengthwise with a sharp blade and moved to a 25 cm^3^ culture plate containing 5 mL of Dulbecco’s modified Eagle medium (DMEM, Sigma-Aldrich, St. Louis, MO, USA) enriched with 5% fetal bovine serum (FBS; Gibco, Carlsbad, CA, USA) and 1% penicillin-streptomycin (Pen-Strep; Sigma-Aldrich, St. Louis, MO, USA). To detach the epithelial cells from the serous cystadenoma, the plate was gently shaken for 48 h at 25 °C to avoid stromal cell contamination. Subsequently, the isolated morphologically epithelial-like cells were picked up using a pipette and then cultured in a humidified incubator containing 5% CO_2_ at 37 °C; the culture medium was replaced every 2–3 days. After 7 days, when the cells attained ~30% confluency, the medium was replenished every 2–3 days until the cells reached 60–70% confluency.

### 2.2. Viral Vector Construction and Cell Transfection

Using the Gateway system (Invitrogen, Carlsbad, CA, USA) [13,14], lentiviral vector plasmids were created to generate immortalized cells. An earlier report showed that activated CDK4 (CDK4^R24C^), human cyclin D1, and hTERT are essential for immortalizing epithelial cells [15]. Dr. E Hara (The Cancer Institute of JFCR, Tokyo, Japan) created genes encoding hTERT, human cyclin D1, and inhibitor-resistant human CDK4 (CDK4^R24C^) that were first inserted into entry vectors using the BP reaction (Invitrogen, Carlsbad, CA, USA) and subsequently inserted into the lentiviral vector CSII-CMV-RfA, which was a gift from Dr. H Miyoshi (RIKEN Bio-Resource Center, Tsukuba, Japan). In this way, *cyclin D1*, CSII-CMV-*hTERT*, and *hCDK4R24C* vectors were generated. As previously described, recombinant lentiviruses with vesicular stomatitis virus G glycoprotein were produced [16]. Dr. Goto (Aichi Cancer Research Institute, Nagoya, Japan) gifted cDNAs encoding mutant *KRAS (KRAS^v12^)* and *PIK3CA (PIK3CA^E545K^)*. Moreover, lentiviral infection of human mutant *KRAS* and mutant *PIK3CA* expression vectors (pCMSCV-EM7bsd-KRAS and pCMSCV-EM7bsd-PIK3CA, respectively) was performed to establish *KRAS* and *PIK3CA* mutant-overexpressing cells. The resulting cell lines were designated as HOVs-cyst-1 (wildtype), HOVs-cyst-1*KRAS* (*KRAS* mutant), HOVs-cyst-1*PIK3CA* (*PIK3CA* mutant), and HOVs-cyst-1*KRAS + PIK3CA* (*KRAS + PIK3CA* mutation concurrent).

### 2.3. Cell Culture and Cystadenoma Cell Lines

Immortalized human ovarian cystadenoma epithelial HOVs-cyst-1, HOVs-cyst-1*KRAS*, HOVs-cyst-1*PIK3CA*, and HOVs-cyst-1*KRAS + PIK3CA* cells were initially established from ovarian serous cystadenoma epithelial cells that have been cultured in a controlled environment at 37 °C, with a 5% CO_2_ atmosphere, in F-medium enriched with 5% FBS (Gibco, Carlsbad, CA, USA) and 1% penicillin-streptomycin (Pen-Strep; Sigma-Aldrich, St. Louis, MO, USA).

### 2.4. Population Doubling Assay

Cells (1 × 10^5^ cells/mL) were seeded in 25 cm^3^ culture plates and grown to approximately 80% confluence before passaging. The population doubling level was determined using the following formula:Population doubling level (PDL) = log_2_(a/b)
where “a” represents the estimate of cells calculated in the passage, and “b” represents the number of cells implanted [17].

### 2.5. Whole-Exome Profiling

To detect any germline or inherent mutations in the HOVs-cyst-1 cells, whole-exome profiling was performed as previously described [18]. First character discipline was determined by generating a DNA integrity number using the Agilent 2000 TapeStation (Agilent Technologies, Santa Clara, CA, USA). Thereafter, whole-exome sequencing was performed with intensified amplicon using the Illumina MiSeq sequencing platform (Illumina, San Diego, CA, USA). The Genome Jack bioinformatics pipeline (Mitsubishi Space Software, Tokyo, Japan) was then used to analyze the sequencing data. Sequence alignment, variant calling, variant filtering, variant annotation, and variant prioritization were subsequently performed to ensure accurate reporting of analytical sensitivity and specificity over numerous steps.

### 2.6. Immunocytochemistry (ICC)

To confirm the epithelial origin and expression pattern, immortalized cultured cells (HOVs-cyst-1) were placed on Lab-Tek chamber slides (Thermo Fisher Scientific, Waltham, MA, USA) for 24 h. The cells were fixed with 4% formalin, permeabilized with 0.1% Triton, and subsequently incubated overnight at 4 °C with the primary antibody pan-cytokeratin, PAX 8, vimentin, p53, ER, and PR (Appendix A). This was followed by washing with PBS. The secondary antibody was then added, and the cells were incubated for 1 h at room temperature (25 °C) and visualized using a Histofine SAB-PO kit (Nichirei, Tokyo, Japan).

### 2.7. Western Blot Analysis

Cell pellets were lysed in the lysis buffer. The samples were then heated for 10 min at 70 °C and subsequently kept on ice for 1 min. LDS buffer and sample-reducing buffer were then added, and the samples were centrifuged for 5 min at 150 rpm. Duplicate samples (10 μg protein concentration in each) were then subjected to sodium dodecyl sulfate (SDS)-polyacrylamide gel electrophoresis (Invitrogen, Carlsbad, CA, USA) and transferred to polyvinylidene fluoride (PVDF) membranes using Bio-Rad semi-dry trans-blotters (Trans-Blot^®^ SD cell, USA). The membranes were then blocked in LI-COR blocking buffer (LI-COR, Lincoln, NE, USA) for 1 h at room temperature (25 °C). Primary antibodies (Appendix A and Appendix A) were then added, and the membranes were incubated overnight at 4 °C on a shaker. The membranes were washed four times for 5 min each with TBST and treated with secondary antibodies (goat anti-mouse or goat anti-rabbit IR-Dye 670- or 800 CW labeled) for 1 h at room temperature. The probed membranes were then washed with TBST and imaged using an LI-COR Odyssey scanner (LI-Odyssey Infrared Imaging System, ICW, Lincoln, NE, USA). The Odyssey 3.0 analytical software (Model-9120, S/N: ODY-2280, LI-COR, Lincoln, NE, USA) was used to determine the raw intensity and near-infrared fluorescence values; intra-lane background signals were eliminated, and boxes were manually regulated over each band of interest.

### 2.8. Cell Proliferation Assay 

HOVs-Cyst-1 cell proliferation was determined in a 0.5% growth medium via the MTT assay [19]. Briefly, the cells were seeded in 96-well plates (3000 cells/well) and subjected to MTT assay. The results were expressed as mean ± standard deviation (SD) based on data obtained for experiments conducted in triplicate.

### 2.9. Wound Healing Assays

Cells were seeded in 6-well culture plates at a density of 1 × 10^6^ cells per well and allowed to grow until 90–100% confluency. A sterile pipette tip was then used to create a straight scratch in the cell layer. All floating cells were removed, and the cell layer was washed twice with the culture medium. Twenty-four hours after scraping, the rate of defective outcome was determined.

### 2.10. Matrigel Invasion Assay

Corning BioCoat Matrigel Invasion Chamber (Discovery Labware Inc., Bedford, MA, USA) was used for the invasion assay under controlled conditions. The pore size of the chamber was 8 μm. Following the addition of 500 µL of serum-free medium to the upper and bottom chambers, the chamber was incubated in a humidified tissue culture incubator at 37 °C and 5% CO_2_ for 1–2 h. Before the cells were plated in the upper chamber, the serum-free medium was taken out from the lower and upper chambers. The lower compartment of the 24-well plate was then filled with 900 µL of DMEM with 20% FBS as a chemoattractant. The upper chamber was used for seeding of cells at a density of 25,000/350 µL in a serum-free medium. The plate was then incubated for 24 h. Next, the medium was removed from both the chambers and sterile PBS was added twice to wash the chambers. The Matrigel membranes were then fixed with 3.7% paraformaldehyde for 2 min, permeabilized with 100% methanol for 20 min, and stained with Giemsa for 15 min. Between each step, the cells were washed twice with PBS, after which the uninvaded cells were carefully removed using a cotton swab, and the number of migrating cells was counted in fifteen non-overlapping fields using light microscopy (BX41, Olympus, Tokyo, Japan).

### 2.11. Anchorage-Independent Assay

Cells were seeded, at a density of 10,000 cells/well, in a 24-well plate containing a top agar layer enriched with 2X-DMEM with 0.33% Nobel agar and 5% FBS; a lower layer was prepared using 2X-DMEM containing 0.5% agar and 5% FBS. Next, a culture medium (1 mL) was added after solidification and incubated at 37 °C for 3 weeks. Colonies wider than 0.05 mm were counted. The cell line, SKOV3, was used as a positive control; these cells formed colonies within 10 days of embedding.

### 2.12. Nude Mouse Xenograft Experiments

The mouse xenograft experiment was performed using 4-week-old female athymic BLAB/c nu/nu mice (Charles River Japan Inc., Kanagawa, Japan). Each experimental group comprised six mice. Cultured cells (2.5 × 10^7^ cells/mL) were injected subcutaneously and intraperitoneally on the left flank. Tumor growth was investigated over a couple of months or until the death of mice. For the confirmation of a non-transformed phenotype, mice inoculated with immortalized HOVs-cyst-1 cells were observed for 10 months along with the other injected mice.

### 2.13. Immunohistochemistry (IHC)

Immunohistochemistry of mouse xenograft tumor was performed on paraffin-embedded samples. Briefly, paraffin-embedded tissues were sectioned serially at a thickness of 5 μm. Few sections were stained with hematoxylin and eosin for histological evaluation and others were used for IHC. Immunohistochemistry was done on deparaffinized sections, which were incubated overnight with pan-cytokeratin, PAX8, vimentin, p53, ER, or PR at 4 °C (Appendix A). Antigen retrieval was performed using sodium citrate buffer (pH 6, Ref-S1699, Dako, CA, USA). Samples were examined under a light microscope by a pathologist, who was blinded to the clinicopathological factors.

### 2.14. Statistical Analysis

Data were expressed as the mean ± SD based on triplicate experiments. Statistical analyses were performed via Student’s *t*-test using the SPSS software (version 21, IBM, Armonk, NY, USA), and a *p*-value < 0.05 was considered statistically significant.

## 3. Results

### 3.1. Development of Immortalized Serous Cystadenoma Epithelial Cell Lines

Purified epithelial cells that formed ovarian serous cystadenoma were successfully immortalized by combinatorial transfection of *CDK4*(*CDK4^R24C^*), human *cyclinD1*, and humane telomerase reverse transcriptase (*hTERT*) genes. The primary and immortalized cystadenoma epithelial HOVs-cyst-1 cells showed no framework-related differences (Figure 1b), indicating that our immortalized serous cystadenoma epithelial cells were pure and non-transmutated and did not dedifferentiate following prolonged cultivation (Figure 1c). Mutant cell lines also showed the same phenotypes as the immortalized cystadenoma epithelial HOVs-cyst-1 cells. We named these cell lines as HOVs-cyst-1 (wild-type), HOVs-cyst-1*KRAS* mutant, HOVs-cyst-1*PIK3CA* mutant, and HOVs-cyst-1(*KRAS + PIK3CA*) mutant cell lines.

### 3.2. Immunocytochemical (ICC) and Western Blot Expression Pattern in Immortalized Serous Cystadenoma Epithelial Cell Lines

ICC and western blot analyses using an epithelial marker (cytokeratin) were first performed to establish the epithelial ancestry of the immortal serous cystadenoma cells. In addition, stromal and Müllerian-derived markers, vimentin and PAX8, were probed to evaluate stromal cell contamination. Besides, ICC for p53, ER, and PR was also performed to determine the expression of hormone receptors. As expected, all adherent cells expressed cytokeratin, PAX8 but not p53, ER, and PR (Figure 2a, Appendix A). Our results show that the immortalized HOVs-cyst-1 cells were of Müllerian and epithelial origin. Furthermore, HOVs-cyst-1 cells were positive for the expression of both cytokeratin and vimentin, suggesting that some of these cells may have undergone epithelial-mesenchymal transition. Subsequently, we concentrated on the status of *KRAS* and *PIK3CA* mutations. In this regard, western blot analysis revealed that the HOVs-cyst-1*KRAS* mutant cells were characterized by the expression of the RAS/ERK signaling pathway proteins, whereas the HOVs-cyst-1*PIK3CA* mutant cells highly expressed members of the PI3K/AKT signaling pathway. Furthermore, the HOVs-cyst-1*KRAS + PIK3CA* mutant cells exhibited activation of both the RAS/ERK and PI3K/AKT signaling pathways (Figure 2b).

### 3.3. Proliferation, Migration, Invasion, and Anchorage-Independent Assays of a Series of HOVs-cyst-1 Mutant Cells

To analyze the biological and functional behavior of the mutant cells, we conducted an in vitro cell proliferation assay, Matrigel invasion assay, and wound healing assay using the immortalized cells. The proliferation of HOVs-cyst-1 cells with both *KRAS* and *PIK3CA* mutations was higher than that of HOVs-cyst-1 cells with either *KRAS* or *PIK3CA* mutation (*p* < 0.01; Figure 3a). Furthermore, analysis of the migration ability of the cells via an in vitro wound-healing assay showed that immortalized HOVs-cyst-1 cells, with both *KRAS* and *PIK3CA* mutations, had significantly (*p* < 0.01) higher migration abilities than HOVs-cyst-1 cells, with either *KRAS* or *PIK3CA* mutation (Figure 3b,c). The Matrigel invasion assay also revealed that *PIK3CA* mutant HOVs-cyst-1 cells and HOVs-cyst-1 cells, with both *KRAS* and *PIK3CA* mutations, showed higher invasion abilities than the other cells (Figure 3d,e). Statistical analysis showed that the proliferative, migratory, and invasive abilities of HOVs-cyst-1 cells, with both *KRAS* and *PIK3CA* mutations, were significantly higher than the corresponding abilities of the HOVs-cyst-1 cells (*p* < 0.01).

An anchorage-independent assay was also conducted to evaluate the transformation phenotype of the cell lines. HOVs-cyst-1 cells did not develop any colonies, whereas the double *KRAS* and *PIK3CA* mutant HOVs-cyst-1 cells, as well as HOVs-cyst-1 cells, with either *KRAS* or *PIK3CA* mutant, showed a small number of colonies compared with the SKOV3 cells (Figure 4a,b). The absence of cell colonies in anchorage-independent assays with HOVs-cyst-1 cells indicated that immortalized serous cystadenoma epithelial cells did not exhibit transformation features.

### 3.4. Tumorigenic Effect in HOVs-cyst-1 Both KRAS and PIK3CA Mutant Cells

Next, we examined the tumorigenic potential of the series of HOVs-cyst-1 cells in a mouse xenograft model. We observed that only double mutant *KRAS* and *PIK3CA* mutant HOVs-cyst-1 cells developed into massive ascites in nude mice as well as into tumors in the peritoneum (6/6, 100%) (Figure 5a,b). Conversely, the other cells (wild-type HOVs-cyst-1 cells or HOVs-cyst-1 cells, with either *KRAS* or *PIK3CA* mutation) did not induce tumor development in nude mice. Additionally, tumors in the intraperitonium were extremely dispersed throughout the peritoneal organs, digestive tract, liver, and colorectal regions of mice xenografted with double mutant HOVs-cyst-1 cells. Histologically, the xenograft tumors, with double mutant HOVs-cyst-1 cells, further revealed a micropapillary invasive pattern of LGSOC with low nuclear atypia without alveoli. Moreover, the activation of both the RAS/ERK and PIK3CA/AKT signaling pathways in xenograft tumors, with double mutant HOVs-cyst-1 cells, was confirmed via western blot analysis. The expression of phospho-MAPK and phospho-AKT was also confirmed in these cells. Besides, the tumors exhibited diffuse expression of cytokeratin, PAX8, and vimentin but not of p53, ER, and PR in immunohistochemistry, which shows that the expression patterns of these markers were the same as in HOVs-cyst-1 cells (Appendix A).

### 3.5. Evaluation of Other Genetic Mutations in Immortalized HOVs-cyst-1 Cells

To identify any pre-existing somatic mutations of immortal HOVs-cyst-1 cells, we also did whole-exome profiling. However, no cancer-specific alterations, including insertions or deletions, were observed in somatic genes. Furthermore, no copy number alterations in oncogenes, mismatch repair genes, homologous recombination genes, or tumor suppressor genes were observed (Appendix A and Appendix A).

## 4. Discussion

LGSOCs are chemoresistant tumors that account for approximately 10% of serous ovarian carcinomas [20]. Sequencing analysis performed in previous studies showed that LGSOCs are characterized by very few point mutations [8]. Additionally, recent molecular analyses revealed that LGSOCs show a comparatively lower somatic mutation burden, including recurrent mutations in *KRAS* (~22%), *BRAF* (~16%), and *NRAS* (~24%) [8,21,22]. *KRAS* (16–54%) and *BRAF* (2–33%) mutations, which are involved in the ERK pathway, are commonly responsible for the progression of LGSOCs in western countries [7,8,9], whereas the highest frequency of *PIK3CA* (60%) oncogenic mutations, linked to the AKT/mTOR pathway, has been observed in the Japanese population [12]. To understand the etiology of LGSOCs, we successfully established a novel in vitro model of LGSOCs. The experimental HOVs-cyst-1 cells were first established from serous cystadenoma epithelial cells of the ovary without any existing carcinomas and then immortalized by overexpression of cyclin D1, CDK4, and hTERT. Previous reports have shown that immortalized human ovarian surface epithelial cells (OSE) and fallopian tube secretory epithelial cells (FTSECs) generated by transduction of a core set of genes, *CDK4* and *cyclin D1* in combination with *hTERT*, did not induce major chromosomal alterations or transformed phonotype [23,24]. We evaluated immunophenotypic features of immortalized HOVs-cyst-1 cells and found diffuse expression of pan-cytokeratin and PAX8. PAX 8 is a transcription factor reported to be a relatively specific marker for Müllerian tumors. Previously, PAX8 expression was detected in 77% (23/30) of serous cystadenomas using immunohistochemistry [25]. Together with this report, the results of the present study suggest that serous cystadenomas might be originated from Müllerian epithelium. Subsequently, we introduced *KRAS* or *PIK3CA* mutations separately or simultaneously in HOVs-cyst-1 cells. We observed that HOVs-cyst-1 cells, with both *KRAS* and *PIK3CA* mutations, showed significantly higher cell proliferation and migration ability compared with HOVs-cyst-1 cells, with single *KRAS* or *PIK3CA* mutation. The evaluation of the phenotype of transformed HOVs-cyst-1 cells via an anchorage-independent assay showed that cells transformed with single *KRAS* or *PIK3CA* mutants*,* as well as those transformed with the double *KRAS*/*PIK3CA* mutant, exhibited similar colony formation numbers. Furthermore, the results of our in vitro experiments indicated that the oncogenic potential was higher in double oncogenic mutant HOVs-cyst-1 cells than in single oncogenic mutant HOVs-cyst-1 cells. A previous study on colon cancer showed that the concurrence of *KRAS* and *PIK3CA* mutations in cells induced the potential synergistic hyperactivation of the RAS/ERK and PI3K/AKT pathways, which result in uncontrolled cell proliferation and metastasis [26]. Taken together, the findings of this study, as well as those of previous studies, suggest that activation of the RAS/ERK and PI3K/AKT signaling pathways confers a substantial growth advantage to normal epithelial cells.

Additionally, examination of the carcinogenic potentials of *KRAS* and *PIK3CA* mutations using a xenograft model revealed that HOVs-cyst-1 cells, with double *KRAS* and *PIK3CA* mutations, exhibited tumorigenic potential, suggesting that simultaneous activation of the KRAS/ERK and PIK3CA/AKT signaling pathways may play an essential role in LGCS carcinogenesis. Similar phenomena have been demonstrated in carcinogenic models of breast carcinoma [27]. Additionally, in this study, HOV-cyst-1 cells, as well as either *KRAS* or *PIK3CA* mutant HOV-cyst-1 cells, exhibited no signs of tumorigenicity. Our recent study involving the use of immortalized endometriotic epithelial cells with either *KRAS* or *PIK3CA* mutations in a mouse xenograft model also showed insufficient tumorigenic potential [28]. Interestingly, the histological examination of double mutant HOVs-cyst-1 in this study showed a micropapillary pattern with small, uniform papillae with cores containing little or no stroma or a complex pattern of elongated, thin papillae lined with serous epithelial cells. Moreover, the tumors exhibited diffuse pan-cytokeratin, as well as PAX8, expression. In a previous study, immunohistochemistry was carried out using PAX8 on 102 low-grade serous neoplasms of the ovary, including both SBTs and well-differentiated serous carcinoma, and PAX8 staining was seen in 100% samples [29]. Taken together, our results prove that the cells were LGSOC cells of Müllerian epithelium. Current xenograft and histological results have indicated that serous cystadenomas are a percussor of LGSOCs. To the best of our knowledge, this study is the first to report an in vitro carcinogenesis model for LGSOCs.

In the present study, two genetic mutations (*KRAS + PIK3CA)* were sufficient to induce LGSOC tumor growth and progression in nude mice. Previously, we established an in vitro stepwise carcinogenic model of HGSCs using primary fallopian tube secretory epithelial cells and found that at least three successive mutations are necessary to drive tumorigenesis [26]. Using a mathematical approach to analyze genome-wide sequencing data, Vogelstein et al. also concluded that three genetic alterations are essential to drive the tumorigenesis of the human colon and lung [30,31]. These findings suggest that at least three oncogenic alterations are required for carcinogenesis from normal epithelial cells to carcinomas. Thus, we speculated that HOV-cyst-1 cells already have oncogenic mutations. However, whole-exome sequencing to identify the presence of oncogenic mutations in HOV-cyst-1 cells showed no such somatic or germline mutations, suggesting that HOV-cyst-1 cells may harbor epigenetic alterations. Therefore, these benign tumors, such as serous cystadenomas, may already have a one-step carcinogenic stage compared to normal fallopian epithelial cells. While the initiation and progression of carcinomas are thought to be genetic, they are also believed to result from epigenetic abnormalities [32]. Previous studies have indicated that 86% of ovarian serous cystadenomas are polyclonal and non-neoplastic, whereas only 14% of serous cystadenomas are clonal [33]. Additionally, *BRAF* and *KRAS* mutations, which characterize serous borderline tumors (SBTs) and LGCSs, are absent in serous cystadenomas. It has also been reported that a small proportion of these cystadenomas become clonal and that *KRAS* or *BRAF* mutations in some of these clonal cystadenomas lead to the development of SBTs, which are the precursors of LGSOCs [33]. This genetic evidence supports the findings of the present study based on the in vitro carcinogenic model of LGSOC. However, epigenetic contributions to LGSOC development remain unclear. Further, epigenetic changes may be involved in each cell division process and possibly affect cell phenotypes. Recently, Siegmund et al. observed that the estimated rate of epigenetic changes is an order of magnitude higher than the estimated rate of genetic changes and could be a major determinant of clonal evolution [34]. Longitudinal transcriptional and genetic analyses of clonal colon cancer cell populations revealed a slowly drifting spectrum of epithelial-to-mesenchymal transcriptional identities that are seemingly independent of genetic variation. DNA methylation landscapes show a correlation with these identities; however, they also reflect an independent clock-like methylation loss process [35]. Taken together, these recent reports, and our current findings based on the in vitro carcinogenic model of LGSOC, suggest that epigenetic changes may contribute to the development of LGSOCs. Therefore, we speculate that epigenetic changes might be an early event in the development of LGSOCs accompanied by genetic alterations. Further studies to clarify the epigenetic changes that are associated with the development of LGSOCs, including DNA methylation, histone modification, nucleosome positioning, and non-coding RNAs, especially microRNA expression, are urgently needed.

There is no specific treatment regimen for LGSOCs. Current treatment strategies for LGSOCs are often unsatisfactory. An in vitro study showed resistance of LGSOCs to CBDCA, PTX, gemcitabine, cyclophosphamide, and cisplatin, and with a lesser probability to doxorubicin etoposide and topotecan [35]. Molecularly targeted agents, especially cyclin-dependent kinase (CDK) and MEK inhibitors, are under investigation. Hormonal therapy might bring clinical benefits to women with LGSOC [36,37,38,39,40]. Recently, we analyzed the ER expression status in LGSOCs, SBTs, and SCAs, and found that the activation of both the ER and PI3K/AKT signaling pathways may play an important role in the LGSOC carcinogenesis (paper submitted). Therefore, we speculate that downregulation of ER using fulvestrant alone, or with a combination of PI3K inhibitor might provide clinical benefits for LGSOC patients. In this study, we successfully established a novel in vitro model of LGSOC from serous cystadenomas and developed carcinoma cell lines by introducing oncogenic *KRAS* and *PIK3CA* mutations. In our ongoing research, we are analyzing the sensitivity of these established cell lines to ER and PI3K inhibitors; the results of these investigations should provide us with valuable information about the therapeutic potential and the possible mechanisms of action of these inhibitors.

Although we have characterized a new xenograft model of human LGSOC for the first time, two major limitations remain. First, the current in vitro model was established based on experiments using two-dimensional (2D) cell cultures; the use of three-dimensional (3D) cell cultures, which mimic the in vivo conditions more closely, is essential to develop more relevant cell models. Second, in the current study, we have developed LGSOC cell lines and transplanted them in nude mice, but the development of genetically engineered mouse models of LGSOC is essential for evaluating specific mutations and understanding the mechanisms underlying the tumorigenesis in the future.

## 5. Conclusions

In this study, we observed that both *KRAS* and *PIK3CA* mutations, that is, activation of the KRAS/ERK and PIK3CA/AKT signaling pathways, play an essential role in the development of LGSOCs. We also successfully developed a stepwise model for in vitro development of LGSOCs using immortalized cystadenoma cells. The fact that carcinogenesis from benign tumors requires only two oncogenic hits, and not three, as observed in this study, suggests that benign tumors are one step up with respect to the three stages of carcinogenesis compared with normal epithelial cells. This stepwise in vitro carcinogenic model of LGSOCs will contribute to further research on the etiopathogenesis of LGSOCs and should facilitate the identification of novel molecular targets and biomarkers for early detection of LGSOCs.

## Figures and Tables

**Figure 1 cancers-14-01506-f001:**
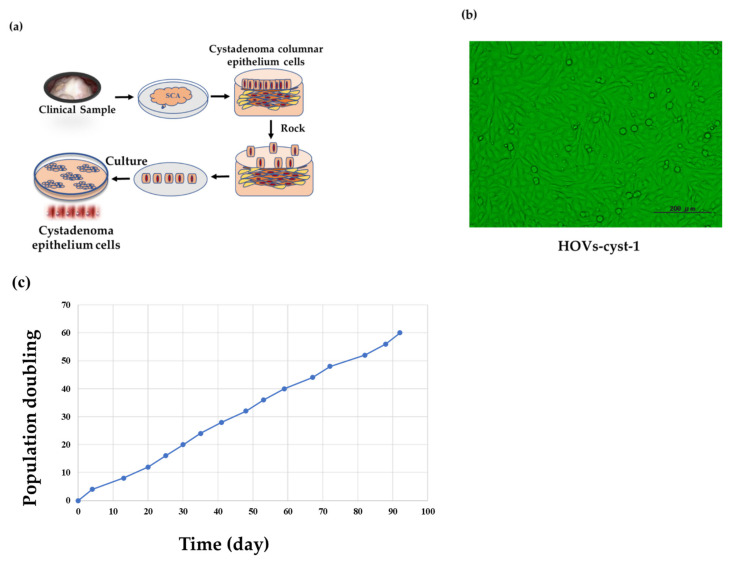
Immortalization of serous cystadenoma epithelial cells (HOVs-cyst-1). (**a**) Tissue acquisition and purification, including dissection, epithelial cell disruption, acquisition of adherent cells, and culturing in DMEM supplemented with FBS (5%) and Pen-Strep (1%). (**b**) Morphological characteristics of immortalized serous cystadenoma epithelial cells. (**c**) Growth curve of immortalized serous cystadenoma epithelial cells (HOVs-cyst-1), as determined via population doubling assays.

**Figure 2 cancers-14-01506-f002:**
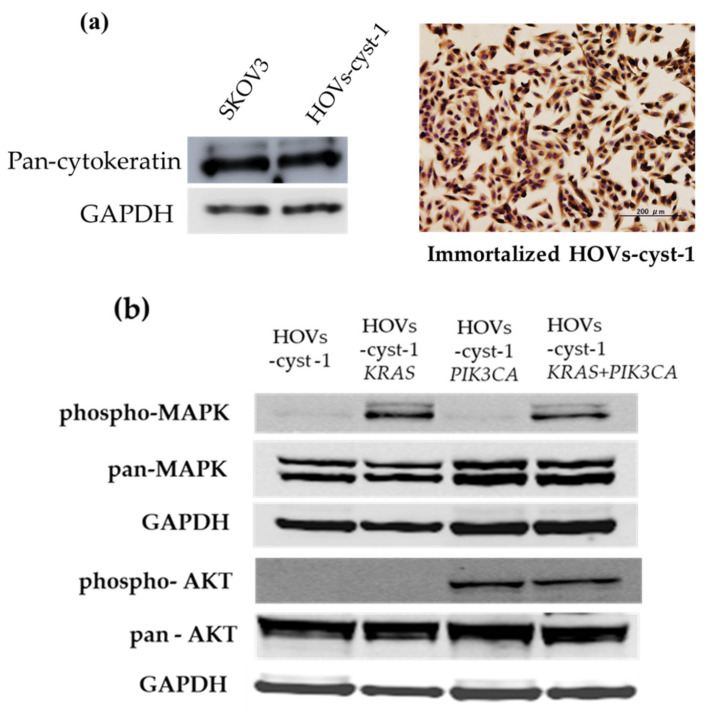
Characterization of HOVs-cyst-1 cells and oncogenes via western blot analysis and immunohistochemistry. (**a**) Western blot analysis and immunohistochemistry of pan-cytokeratin in HOVs-cyst-1 cells. (**b**) Western blot analysis of phospho-MAPK, pan-MAPK, phospho-AKT, and pan-AKT expression in the series of HOVs-cyst-1 cells (original HOVs-cyst-1 cells, HOVs-cyst -1*KRAS* cells, HOVs-cyst-1*PIK3CA* cells, and HOVs-cyst-1*KRAS* + *PIK3CA* cells). Uncropped Western Blots of pan-cytokeratin, phospho-MAPK, pan-MAPK, phospho-AKT, and pan-AKT expression can be found at Appendix A.

**Figure 3 cancers-14-01506-f003:**
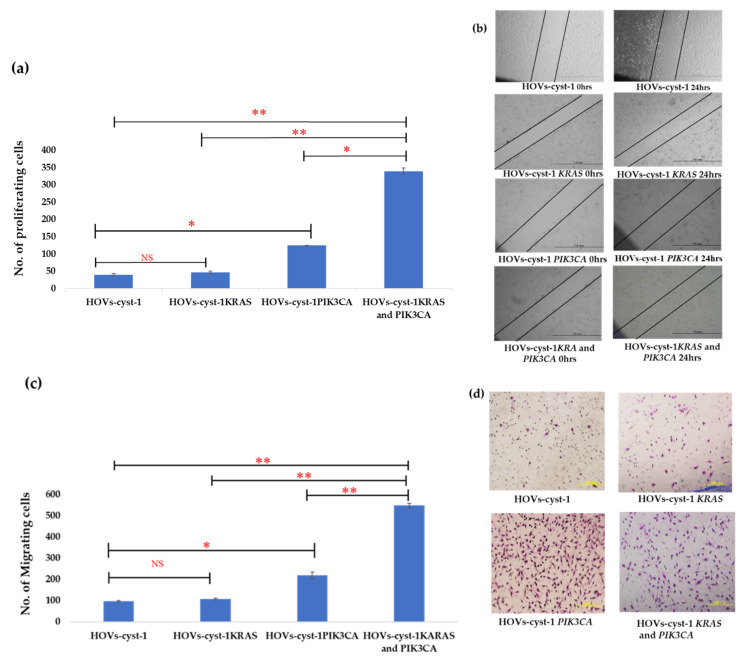
Proliferation of HOVs-cyst-1 cells. (**a**) Proliferation capability of HOVs-cyst-1cells, with both *KRAS* and *PIK3CA* mutations (significantly higher than those of HOVs-cyst-1 cells or HOVs-cyst-1 cells, with either *KRAS* or *PIK3CA* mutation; * *p* < 0.05, ** *p* < 0.01). (**b**,**c**) Migration ability of cells based on wound healing assay. HOVs-cyst-1 cells, with both *KRAS* and *PIK3CA* mutations, showed higher cell migration ability than HOVs-cyst-1 cells or HOVs-cyst-1, with either KRAS or PIK3CA mutation. The photographs were taken immediately after the scratch (0 h, upper left panel) and 24 h after the scratching (upper right panel). (**d**,**e**) Invasion rates of cells based on Matrigel invasion assay. HOVs-cyst-1 cells, with *PIK3CA* mutation, and HOVs-cyst-1 cells, with both KRAS and PIK3CA mutations, showed higher invasion rates than the other cells (* *p* < 0.05, ** *p* < 0.01). The error bar indicates the standard deviation. (NS; no significant difference; * *p* < 0.05, ** *p* < 0.01).

**Figure 4 cancers-14-01506-f004:**
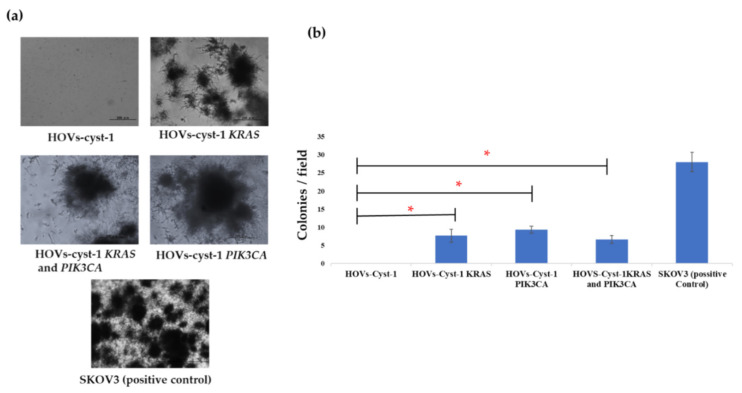
Anchorage-independent growth assay. (**a**) Cell colonies in Matrigel. The HOVs-cyst-1 cells, with different mutations, showed colonies in Matrigel; SKOV3 cells were used as positive control. (**b**) All mutant cells (HOVs-cyst-1*KRAS*, HOVs-cyst-1*PIK3CA*, and HOVs-cyst-1*KRAS* + *PIK3CA*) formed colonies, and the number of colonies did not differ significantly (* *p* < 0.05).

**Figure 5 cancers-14-01506-f005:**
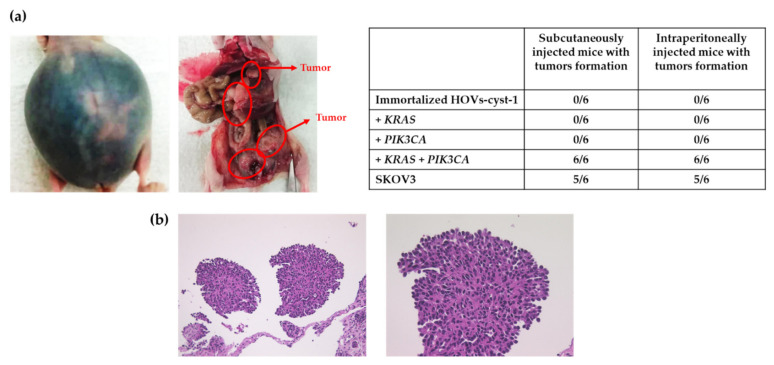
Mouse xenograft assay using HOVs-Cyst-1 cells with genetic mutations (either *KRAS* or *PIK3CA*, or both *KRAS* and *PIK3CA*). (**a**) Intraperitoneal inoculation resulted in tumor formation only when HOVs-1-Cyst-1 cells were characterized by both *KRAS* and *PIK3CA* mutations. The other series of HOVs-Cyst-1 cells did not form tumors in nude mice. (**b**) Hematoxylin and eosin (H & E) staining of HOVs-Cyst-1 cells with both *KRAS* and *PIK3CA* mutations showing a micropapillary LGSOC invasive pattern with low nuclear atypia without alveoli.

## Data Availability

The data presented in this study are available on request from the corresponding author (K.N.).

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
