# Peer review of "Development of Low-Grade Serous Ovarian Carcinoma from Benign Ovarian Serous Cystadenoma Cells"

_cancers, 2022, doi:10.3390/cancers14061506_

Round 1

Reviewer 1 Report

I read with interest the paper entitled “Establishment of low-grade serous ovarian carcinoma from benign ovarian serous cystadenoma cells” by Dey and colleagues. This in vitro study aimed to add information to the current understanding regarding the molecular pathogenesis of low-grade serous ovarian carcinomas, but it may also contribute to the development of novel therapeutic agents for the management of low-grade serous ovarian carcinomas. I think it may be accepted after minor revision.

Title

I suggest changing the title into “Development of low-grade serous ovarian carcinoma from benign ovarian serous cystadenoma cells”

Keywords

I suggest adding “ovarian cancer”

Simple summary

Appropriate

Abstract

Line 36: what’s the meaning of “LGSOCs in nude mice”? Please delete or explain it

Introduction

- The differentiation in type 1 and 2 is for epithelial ovarian cancer and not for serous. Please correct it

- Type 1 includes also endometrioid ovarian cancer. Please add it

- Please add a little more information (1 or 2 sentences) about the clinical behavior of LGSOC

Materials and Methods

- Please add some information about the pathological analysis of the specimen (was it analyzed by a pathologist specifically trained in gynecologic oncology? Which kind of classification refers to?...)

- Please add this information from the Results section to the M&M: “Serous cystadenoma epithelial cells were collected from a patient via laparoscopic 219 bilateral salpingo-oophorectomy. To confirm that malignant lesions did not mix with our 220 collected samples, examined the tissues for pathological outcomes, which showed no ma-221 lignant tissue.”

- Please add this information from the Results section to the M&M: “An earlier report showed that activated CDK4 (CDK4R24C), cyclin D1, and 222 hTERT are essential for immortalizing epithelial cells [19]. Therefore, these genes were 223 transfected into the serous cystadenoma epithelial cells via a lentiviral vector.”

Discussion

- Could you try to discuss a little bit more in detail about the differences between western and Japanese LGSOCs , especially for PIK3CA mutations? Could it be hypothesized another in vitro model? Do they behave differently?

- Are you planning to test drugs on these molecular targets? Could you please add some discussion about the potential impact of your study in this sense?

Figures

Appropriate

Language

The whole paper should be revised by an English native speaker

Author Response

Point-wise responses to the comments made by Reviewer 1:

Comments and Suggestions for Authors

Q1)

I read with interest the paper entitled “Establishment of low-grade serous ovarian carcinoma from benign ovarian serous cystadenoma cells” by Dey and colleagues. This in vitro study aimed to add information to the current understanding regarding the molecular pathogenesis of low-grade serous ovarian carcinomas, but it may also contribute to the development of novel therapeutic agents for the management of low-grade serous ovarian carcinomas. I think it may be accepted after minor revision.

A1)

Thank you for your kind consideration.

Q2) Title

I suggest changing the title into “Development of low-grade serous ovarian carcinoma from benign ovarian serous cystadenoma cells”

A2)

Thank you for suggesting a pertinent change in the title. We have modified it accordingly. Please check the revised title.

Q3) Keywords

I suggest adding “ovarian cancer”

A3)

As suggested, we have added “ovarian cancer” in the list of keywords. Please check the keywords.

Q4) Simple summary

Appropriate

A4)

We thank you for the comment.

Q5) Abstract

Line 36: what’s the meaning of “LGSOCs in nude mice”? Please delete or explain it

A5)

We apologize for this mistake. We have deleted it. Please see the revised abstract (line 36).

Q6) Introduction

  1. The differentiation in type 1 and 2 is for epithelial ovarian cancer and not for serous. Please correct it
  2. Type 1 includes also endometrioid ovarian cancer. Please add it
  3. Please add a little more information (1 or 2 sentences) about the clinical behavior of LGSOC

A6)

  1. Thanks a lot for your valuable advice. We have made the suggested

 correction. Please see the introduction section (line 44–45).

  1. Thank you for this suggestion. We have included endometrioid ovarian cancer. Please see the introduction section (line 50).
  2. We have added more information about the clinical behavior of LGSOC. Please see the introduction section (line 52).

Q7) Materials and Methods

  1. Please add some information about the pathological analysis of the specimen (was it analyzed by a pathologist specifically trained in gynecologic oncology? Which kind of classification refers to?...)
  2. Please add this information from the Results section to the M&M: “Serous cystadenoma epithelial cells were collected from a patient via laparoscopic 219 bilateral salpingo-oophorectomy. To confirm that malignant lesions did not mix with our 220 collected samples, examined the tissues for pathological outcomes, which showed no ma-221 lignant tissue.”
  3. Please add this information from the Results section to the M&M: “An earlier report showed that activated CDK4 (CDK4R24C), cyclin D1, and 222 hTERT are essential for immortalizing epithelial cells [19]. Therefore, these genes were 223 transfected into the serous cystadenoma epithelial cells via a lentiviral vector.”

A7)

  1. Thank you for the comments. We have added information related to the pathologist. Please check the materials and method section (page 2, line 92–93).
  2. Thank you for the suggestions. We have relocated the relevant text from the results section to the materials and method section. Please see the materials and methods section (page 3, line 96–98 and 113–115).

Q8) Discussion

  1. - Could you try to discuss a little bit more in detail about the differences between western and Japanese LGSOCs, especially for PIK3CA mutations?
  2. Could it be hypothesized another in vitro model? Do they behave differently?
  3. Are you planning to test drugs on these molecular targets? Could you please add some discussion about the potential impact of your study in this sense?

A8)

  1. According to your suggestion, we have added more information related to western and Japanese LGSOCs. Please see the discussion section (page 12, line392-395).
  2. Another in vitro model, such as KRAS + additional oncogene (ERBB2) or PIK3CA + additional oncogenes (ERBB2), might be possible. We are planning to undertake this investigation in the future. As such, presently, we cannot comment on whether another model would behave differently.
  3. We are conducting experiments to test drugs on these molecular targets. We have added the information sought by you in the discussion section (page 13, line 483-499).

Q9) Figures

Appropriate

A9)

Thank you for the positive comment.

Q10) Language

The whole paper should be revised by an English native speaker

A10)

We have got the entire revised manuscript edited by a native English speaker.

Reviewer 2 Report

This article entitled “Establishment of low-grade serous ovarian carcinoma from benign ovarian serous cystadenoma cells” is constructive. The authors established a node mouse xenograft model; the results of tumorigenic potential of the series of HOVs-cyst-1 cells in a mouse xenograft model showed that only double mutant KRAS and PIK3CA mutant HOVs-cyst-1 cells developed into massive ascites in nude mice as well as tumors in the peritoneum. This issue, from benign ovarian serous cystadenoma transforming to low-grade serous ovarian carcinoma, is critical important for clinical practice and attracts the readers. I suggest accept the article.

Author Response

Point-wise response to the comment made by Reviewer 2:

Comments and Suggestions for Authors

Q1)

This article entitled “Establishment of low-grade serous ovarian carcinoma from benign ovarian serous cystadenoma cells” is constructive. The authors established a node mouse xenograft model; the results of tumorigenic potential of the series of HOVs-cyst-1 cells in a mouse xenograft model showed that only double mutant KRAS and PIK3CA mutant HOVs-cyst-1 cells developed into massive ascites in nude mice as well as tumors in the peritoneum. This issue, from benign ovarian serous cystadenoma transforming to low-grade serous ovarian carcinoma, is critical important for clinical practice and attracts the readers. I suggest accept the article.

A1)

We appreciate your positive evaluation of our manuscript and thank you for recommending its publication.

Reviewer 3 Report

In this original article, Dey et al. reported how they established a low-grade serous ovarian carcinoma cell line from benign ovarian serous cystadenoma cells. They used oncogenic KRAS and PIK3CA gene mutation transfer into the benign cells. 

The introduction is correct.

In the material and methods, it should be cleared when were the benign ovarian cells isolated from the human ovarian surgical sample. The ethical approval was prepared in 2007. How long was the benign cell line maintained before the transfer of the malignant gene mutations?

The results are clear. How stable was the new malignant cell line? Was it observed?

The discussion is also clear.

After minor revision, I suggest accepting the manuscript for publication. 

Author Response

Point-wise responses to the comments made by Reviewer 3:

Comments and Suggestions for Authors

Q1)

In this original article, Dey et al. reported how they established a low-grade serous ovarian carcinoma cell line from benign ovarian serous cystadenoma cells. They used oncogenic KRAS and PIK3CA gene mutation transfer into the benign cells. The introduction is correct.

A1)

Thank you for your kind consideration.

Q2)

  1. In the material and methods, it should be cleared when were the benign ovarian cells isolated from the human ovarian surgical sample.
  2. The ethical approval was prepared in 2007.
  3. How long was the benign cell line maintained before the transfer of the malignant gene mutations?

A2)

  1. We thank you for the valuable suggestions. We have mentioned the year when the benign ovarian cells were isolated in the materials and methods section (page 2, line91).
  2. We apologize for this mistake. We have included the updated IRB number in the materials and methods section (page 3, line94).
  3. The benign cell line was maintained for 3 months before the transfer of the malignant gene mutations.

Q3)

The results are clear. How stable was the new malignant cell line? Was it observed?

A3)

Thank you for the pertinent query. As for HOVs-cyst-1 wild type, we checked the malignant cell lines for 3 months. The phenotype of the mutant cell line was the same as that of HOVs-cyst-1 wild type.

Q4)

The discussion is also clear. After minor revision, I suggest accepting the manuscript for publication

A4)

We have made all the suggested revisions. We thank you for recommending the publication of our manuscript.

Reviewer 4 Report

In this article, Dey P et al. reported in vitro carcinogenesis model of LGSOCs by introducing oncogenic KRAS and PIK3CA gene mutations in immortalized HOVs-cyst-1 cells derived from a serous cystadenoma sample. The nude mouse xenograft tumors inoculated with HOVs-cyst-1 cells with KRAS and PIK3CA co-mutations exhibited peritoneal carcinomatosis with massive ascites similar to advanced ovarian cancer clinical cases. Although it is very interesting to develop a LOSOCs carcinogenesis model, there are several points to be addressed before acceptance of this draft.

Major comments

#1. Introduction; Are there any reports that show the rate of LGSOCs with both oncogenic KRAS and PIK3CA gene mutations? Authors describe that KRAS or BRAF mutation is dominant in LGSOC in western countries, while PIK3CA is the most prevalent (60%) in Japan. If the proportion of LGSOC with both oncogenic KRAS and PIK3CA gene mutations is small, the LOSOCs carcinogenesis model developed by authors (HOVs-cyst-1(KRAS+PIK3CA) cells.) is not a suitable model for most LGSOC clinical cases, diminishing the value of this study.

#2. 2. Materials and Methods, 2.1. Purification and Isolation of Ovarian Serous Cystadenoma Epithelial Cells; It is unclear how to purify the epithelial cells of ovarian serous cystadenoma. Was cell sorting or MACS selection performed to collect only the epithelial cells? In addition, it is necessary to show the data representing at least an epithelial marker, a stromal marker, a Mullerian-derived marker (e.g. PAX8), p53, and hormone markers (e.g. ER, PgR) using immunofluorescence or flow cytometry. The expression pattern of the markers in the primary tumor (ovarian serous cystadenoma) are also needed to be included.

#3. Figure 5; Immunohistochemical analysis using the xenograft tumors of HOVs-cyst-1(KRAS+PIK3CA) should be added to show the authenticity as LGSOC like the major comment #2 described above.

Minor comments

#1. Line 34-36; It is better to change the sentence as “Furthermore, the transformation of HOV-cyst-1 cells with KRAS and PIK3CA mutations resulted in the development of tumors in nude mice that were grossly and histologically similar to human LGSOCs.”

#2. Line 88-89; Is a 53-year-old woman pre-menopause or post-menopause?

#3. Line 118-119; Please describe which name corresponds to which cell.

#4. Line 145-152; How much protein did you apply in a lane during electrophoresis?

#5. Line 168-169; Please add the pH of sodium citrate buffer and how long the slides were incubated with pan-cytokeratin and GAPDH primary antibodies.

#6. Line 206; Is “Node Mouse Xenograft Experiments” mean “Nude Mouse Xenograft Experiments”?

#7. Line 224-228; It is better to add the picture of the primary cystadenoma epithelial cells before immortalization transduction.

#8. Figure 1(a); What do “SCA” and “Rock” stand for in figure 1(a)?

#9. Line 263-270; To avoid the confusion of readers, this sentence should focus on wound healing assay but not Matrigel invasion assay. The result of Matrigel invasion assay is described in Line 267-270. Also, is “€” in Line 270 mean “e”?

#10. Figure 3a and 3c; Please add the statistical analysis result between HOVs-cyst-1 and HOVs-cyst-1PIK3CA in the figures.

#11. Figure 3(b); It is better to replace the pictures of HOVs-cyst-1KRAS 24hrs, HOVs-cyst-1PIK3CA 24hrs, and HOVs-cyst-1KRAS and PIK3CA 24hrs with higher resolution photos.

#12. Figure 3e; Why do you think the HOVs-cyst-1PIK3CA cells significantly showed higher invasion rates than HOVs-cyst-1PKRAS and PIK3CA? Are there similar publications in the past?

#13. Line 294-298; “Except for the original HOCs-cyst-1 cells and SKOV3 cells (positive control), the HOVs-cyst-1 cells with different mutations 295 showed colonies in Matrigel.” is misleading. SKOV3 cells also showed colonies.

#14. Figure 4b; Please clarify the *bar range which seems to be located between HOVs-cyst-1 and HOVs-cyst-1KRAS and PIK3CA. Also, it is needed to add the statistical analysis results between HOVs-cyst-1 and HOVs-cyst-1 KRAS as well as HOVs-cyst-1 and HOVs-cyst-1 PIK3CA in the figures.

#15. Figure 5; Why do authors think that SKOV3 as positive control did not develop xenograft tumors in all mice?

#16. 3.5. Evaluation of Other Genetic Mutations in Immortalized HOVs-cyst-1 cells; The genomics of primary tumor (ovarian serous cystadenoma) should be evaluated in case the genome profile of immortal HOVs-cyst-1 might be altered after a series of passages. Also, please add the passage number of immortal HOVs-cyst-1 cell submitted to WES.

#17. Discussion; Please mention the information about LGSOC cell line models and the genetically engineered mouse model of LGSOC.

#18. Discussion; The induced genomic changes of hTERT, cyclin D1, and CDK4R24C in the immortalized HOVs-cyst-1 cell are also considered as the potential factors of tumorigenesis in nude mice.

#19. Discussion; Please describe the limitations of this study.

#20. English proof with a native speaker is needed. At least, many spelling/grammatical errors are found below.

 Line 95; move→moved

Line 118, 229, 275, 295; HOVs-cyat-1PIK3CA→HOVs-cyst-1PIK3CA 

Line 123; has→have

Line 142; was→were

Line 216; Were→was

Line 276; HOVs-cyst 1→HOVs-cyst-1

Line 296; HOVs-Cyst-1PIK3CA →HOVs-cyst-1PIK3CA

Line 421; assessed→were assessed

Author Response

Point-wise responses to the comments made by Reviewer 4:

Comments and Suggestions for Authors

Q1)

In this article, Dey P et al. reported in vitro carcinogenesis model of LGSOCs by introducing oncogenic KRAS and PIK3CA gene mutations in immortalized HOVs-cyst-1 cells derived from a serous cystadenoma sample. The nude mouse xenograft tumors inoculated with HOVs-cyst-1 cells with KRAS and PIK3CA co-mutations exhibited peritoneal carcinomatosis with massive ascites similar to advanced ovarian cancer clinical cases. Although it is very interesting to develop a LOSOCs carcinogenesis model, there are several points to be addressed before acceptance of this draft.

Major comments

#1. Introduction; Are there any reports that show the rate of LGSOCs with both oncogenic KRAS and PIK3CA gene mutations? Authors describe that KRAS or BRAF mutation is dominant in LGSOC in western countries, while PIK3CA is the most prevalent (60%) in Japan. If the proportion of LGSOC with both oncogenic KRAS and PIK3CA gene mutations is small, the LOSOCs carcinogenesis model developed by authors (HOVs-cyst-1(KRAS+PIK3CA) cells.) is not a suitable model for most LGSOC clinical cases, diminishing the value of this study.

A1) We thank you for the valuable comments. There are some reports wherein both oncogenic KRAS and PIK3CA or oncogenic BRAF and PIK3CA gene mutations are reported to exist in SBTs (precursor of LGSOCs) or LGSOCs. We have included this information in the introduction section (page 2, line 71–72).

#2. 2. Materials and Methods, 2.1. Purification and Isolation of Ovarian Serous Cystadenoma Epithelial Cells; It is unclear how to purify the epithelial cells of ovarian serous cystadenoma. Was cell sorting or MACS selection performed to collect only the epithelial cells? In addition, it is necessary to show the data representing at least an epithelial marker, a stromal marker, a Mullerian-derived marker (e.g., PAX8), p53, and hormone markers (e.g., ER, PgR) using immunofluorescence or flow cytometry. The expression pattern of the markers in the primary tumor (ovarian serous cystadenoma) are also needed to be included.

A2) Thank you for your valuable advice. We picked up morphologically epithelial-like cells using a pipette. We have included this information in the materials and methods section (page 3, line 106-107). As suggested by you, we did the identification of an epithelial marker, a stromal marker, a Mullerian-derived marker (e.g., PAX8), p53, and hormone markers (e.g., ER, PgR) using immunocytochemistry and have added this information in the materials and methods (page 4, line 154–162), results (page 7, line 264–273), and supplementary (Figure S1 and Table S1) sections.

We could not perform flow cytometry or immunofluorescence analysis because we do not have the required equipment.

#3. Figure 5; Immunohistochemical analysis using the xenograft tumors of HOVs-cyst-1(KRAS+PIK3CA) should be added to show the authenticity as LGSOC like the major comment #2 described above.

A3) Thank you for the valuable suggestion. We performed immunohistochemical analysis using the xenograft tumors of HOVs-cyst-1(KRAS+PIK3CA). We have added relevant information in the materials and methods (page 5, line 226–235), results (page 11, line367–370), and supplementary sections (Figure S4).

Minor comments 

#1. Line 34-36; It is better to change the sentence as “Furthermore, the transformation of HOV-cyst-1 cells with KRAS and PIK3CA mutations resulted in the development of tumors in nude mice that were grossly and histologically similar to human LGSOCs.”

A1) We have rephrased the sentence as suggested (page 1, line 35–36).

#2. Line 88-89; Is a 53-year-old woman pre-menopause or post-menopause?

A2) The patient was post-menopausal. We have included this information in the materials and methods section (page 2, line 91).

#3. Line 118-119; Please describe which name corresponds to which cell.

A3) As suggested, we have included the information (page 3, line 126–128).

#4. Line 145-152; How much protein did you apply in a lane during electrophoresis?

A4) We loaded 10 mg protein per lane for electrophoresis. We have added this information in the materials and methods section (page 4, line 167).

#5. Line 168-169; Please add the pH of sodium citrate buffer and how long the slides were incubated with pan-cytokeratin and GAPDH primary antibodies.

A5) We apologize for the missing information. We have added the information in the materials and methods section (page5, line232–233).

#6. Line 206; Is “Node Mouse Xenograft Experiments” mean “Nude Mouse Xenograft Experiments”?

A6) We are sorry for this mistake. We have corrected it in the revised manuscript (page 5, line 217).

#7. Line 224-228; It is better to add the picture of the primary cystadenoma epithelial cells before immortalization transduction.

A7) We appreciate this pertinent suggestion. However, we are sorry to mention that we could not take any picture of primary cystadenoma cells before immortalization.

#8. Figure 1(a); What do “SCA” and “Rock” stand for in figure 1(a)?

A8) In Figure 1(a), “SCA” is the abbreviation for serous cystadenoma and “Rock” has been used for shake.

#9. Line 263-270; To avoid the confusion of readers, this sentence should focus on wound healing assay but not Matrigel invasion assay. The result of Matrigel invasion assay is described in Line 267-270. Also, is “€” in Line 270 mean “e”?

A9) We thank you for your comments and are extremely sorry for our mistake. As suggested, we have corrected the sentences to ensure that there is no confusion for the readers. Please see the revised text (page 8, line 294–300).

#10. Figure 3a and 3c; Please add the statistical analysis result between HOVs-cyst-1 and HOVs-cyst-1PIK3CA in the figures.

A10) We thank you for this suggestion. We have added the results of statistical analysis between HOVs-cyst-1 and HOVs-cyst-1PIK3CA in the figures (Figure 3(a) and 3(c), page 9).

#11. Figure 3(b); It is better to replace the pictures of HOVs-cyst-1KRAS 24hrs, HOVs-cyst-1PIK3CA 24hrs, and HOVs-cyst-1KRAS and PIK3CA 24hrs with higher resolution photos.

A11) We have replaced the pictures of HOVs-cyst-1KRAS and PIK3CA 24 hrs with ones having higher resolution. Please see Figure 3(b) (page 9).

#12. Figure 3e; Why do you think the HOVs-cyst-1PIK3CA cells significantly showed higher invasion rates than HOVs-cyst-1PKRAS and PIK3CA? Are there similar publications in the past?

A12) We are sorry, but we do not have any clear answer for this phenomenon at present. Moreover, we did not find any published report on a similar phenomenon.

#13. Line 294-298; “Except for the original HOCs-cyst-1 cells and SKOV3 cells (positive control), the HOVs-cyst-1 cells with different mutations 295 showed colonies in Matrigel.” is misleading. SKOV3 cells also showed colonies.

A13) We apologize for this mistake. We have rephrased the sentences and have ensured that no misleading information is conveyed. Please see the revised text (page 11, line 349–352).

#14. Figure 4b; Please clarify the *bar range which seems to be located between HOVs-cyst-1 and HOVs-cyst-1KRAS and PIK3CA. Also, it is needed to add the statistical analysis results between HOVs-cyst-1 and HOVs-cyst-1 KRAS as well as HOVs-cyst-1 and HOVs-cyst-1 PIK3CA in the figures.

A14) Thank you for your valuable suggestions. We have corrected the bar range between HOVs-cyst-1 and HOVs-cyst-1KRAS and PIK3CA and have also added the results of statistical analysis between HOVs-cyst-1 and HOVs-cyst-1KRAS as well as HOVs-cyst-1 and HOVs-cyst-1 PIK3CA in Figure 4(b) (page 10).

#15. Figure 5; Why do authors think that SKOV3 as positive control did not develop xenograft tumors in all mice?

A15) Thank you for this question. We believe that this might be due to some technical error.

#16. 3.5. Evaluation of Other Genetic Mutations in Immortalized HOVs-cyst-1 cells; The genomics of primary tumor (ovarian serous cystadenoma) should be evaluated in case the genome profile of immortal HOVs-cyst-1 might be altered after a series of passages. Also, please add the passage number of immortal HOVs-cyst-1 cell submitted to WES.

A16) We agree with the point made by you. We passaged immortal HOVs-cyst-1 cell three times before using them for WES.

#17. Discussion; Please mention the information about LGSOC cell line models and the genetically engineered mouse model of LGSOC.

A17) Thank you for this comment. We have added the information in the discussion section (page 14, line504-508).

#18. Discussion; The induced genomic changes of hTERT, cyclin D1, and CDK4R24C in the immortalized HOVs-cyst-1 cell are also considered as the potential factors of tumorigenesis in nude mice.

A18) We thank you for this comment. We have added this information in the discussion section (page 12, line398-402).

#19. Discussion; Please describe the limitations of this study.

A19) We thank you for this suggestion. We have mentioned the limitations of our study in the revised manuscript (page 14, line500-508).

#20. English proof with a native speaker is needed. At least, many spelling/grammatical errors are found below.

A20) We have got the revised manuscript edited by a native English speaker.

Q21)

 Line 95; move→moved

Line 118, 229, 275, 295; HOVs-cyat-1PIK3CA→HOVs-cyst-1PIK3CA 

Line 123; has→have

Line 142; was→were

Line 216; Were→was

Line 276; HOVs-cyst 1→HOVs-cyst-1

Line 296; HOVs-Cyst-1PIK3CA →HOVs-cyst-1PIK3CA

Line 421; assessed→were assessed

A21) We apologize for these typographical errors and have corrected them in the revised manuscript (lines 101, 127, 132, 306,152, 240, 252, 528).